# *Auricularia auricula* Melanin Protects against Alcoholic Liver Injury and Modulates Intestinal Microbiota Composition in Mice Exposed to Alcohol Intake

**DOI:** 10.3390/foods10102436

**Published:** 2021-10-14

**Authors:** Yichen Lin, Hua Chen, Yingjia Cao, Yuanhui Zhang, Wenfeng Li, Weiling Guo, Xucong Lv, Pingfan Rao, Li Ni, Penghu Liu

**Affiliations:** 1National Engineering Research Center of JUNCAO Technology, Fujian Agriculture and Forestry University, Fuzhou 350002, China; 3190515031@fafu.edu.cn (Y.L.); 3190515002@fafu.edu.cn (Y.C.); 7200112062@stu.jiangnan.edu.cn (W.G.); 2Institute of Food Science and Technology, College of Biological Science and Technology, Fuzhou University, Fuzhou 350108, China; 082000436@fzu.edu.cn (Y.Z.); pingfan.rao@gmail.com (P.R.); nili@fzu.edu.cn (L.N.); 3Fujian Province Key Laboratory of Agro-Ecological Processes in Hilly Red Soil, Agricultural Ecology Institute, Fujian Academy of Agriculture Sciences, Fuzhou 350003, China; chenhua-kyc@faas.cn; 4Department of Nutrition and Food Safety, School of Public Health, Fujian Medical University, Fuzhou 350122, China; fmulwf@fjmu.edu.cn; 5School of Food Science and Technology, Jiangnan University, Wuxi 214122, China

**Keywords:** *Auricularia auricula* melanin, alcoholic liver injury, intestinal microbiota, liver metabolome, mRNA expressions

## Abstract

The potential effects of *Auricularia auricula* melanin (AAM) on the intestinal flora and liver metabolome in mice exposed to alcohol intake were investigated for the first time. The results showed that oral administration of AAM significantly reduced the abnormal elevation of serum total triglyceride (TG), cholesterol (TC), low density lipoprotein cholesterol (LDL-C), aspartate aminotransferase (AST) and alanine aminotransferase (ALT), and significantly inhibited hepatic lipid accumulation and steatosis in mice exposed to alcohol intake. Besides, the abnormally high levels of bile acids (BAs) and lactate dehydrogenase (LDH) in the liver of mice with alcohol intake were significantly decreased by AAM intervention, while the hepatic levels of glutathione (GSH) and superoxide dismutase (SOD) were appreciably increased. Compared with the model group, AAM supplementation significantly changed the composition of intestinal flora and up-regulated the levels of *Akkermansia*, *Bifidobacterium*, *Romboutsia*, *Muribaculaceae*, *Lachnospiraceae*_NK4A136_group, etc. Furthermore, liver metabolomics demonstrated that AAM had a significant regulatory effect on the composition of liver metabolites in mice with alcohol intake, especially the metabolites involved in phosphatidylinositol signaling system, ascorbate and aldarate metabolism, starch and sucrose metabolism, galactose metabolism, alpha-linolenic acid metabolism, glycolysis/gluconeogenesis, and biosynthesis of unsaturated fatty acids. At the gene level, AAM treatment regulated the mRNA levels of lipid metabolism and inflammatory response related genes in liver, including *ACC-1*, *FASn*, *CPT-1*, *CD36*, *IFN-γ*, *LDLr and TNF-α*. Conclusively, these findings suggest that AAM has potential beneficial effects on alleviating alcohol-induced liver injury and is expected to become a new functional food ingredient.

## 1. Introduction

Alcohol abuse has harmful impacts on human health by damaging liver metabolism functions [1]. Alcoholic liver disease (ALD), mainly including fatty liver, hepatitis, cirrhosis, hepatocyte necrosis and even liver failure, is usually caused by long-term excessive alcohol consumption and has become an ever-growing health issue worldwide [2]. Although the pathologic mechanism of ALD has not yet been clarified, there are many factors affecting the progress of ALD, such as the type of alcoholic beverages, nutritional status, eating habits and so on. Excessive alcohol intake produces large number of destructive endogenous free radicals, and it is difficult for the human body to eliminate these free radicals in a short time, leading to liver damage, hyperglycemia, hyperlipidemia, and even hypertension. How to prevent the pathological process of ALD has become a major global problem [3]. Therefore, researchers all over the world are committed to finding effective strategies to intervene in the pathological development of ALD. It is of great significance to excavate natural products with strong antioxidant effect from food resources for the prevention of alcoholic liver injury.

*Auricularia auricular*, a traditional Chinese edible and medicinal mushroom, has been widely consumed in Asian countries (mainly China, South Korea, Japan, etc.) for thousands years because of its unique flavor, taste and medicinal value. *A. auricular* is rich in a variety of nutrients and active ingredients, including proteins, polysaccharides, melanin, polyphenols, calcium, iron and vitamins [4]. In recent years, most of the researches were focused on the extraction of *A. auricula* polysaccharides, which had been widely proved to have excellent biological activities of immunoregulation [5], antioxidant [6] and anti-tumor [7] through pharmacological research. As a negatively charged brown macromolecule, *A. auricular melanin* (AAM) has higher edible safety and biological activity compared with chemical synthetic pigments [8]. More and more evidences show that natural melanin has a variety of biological activities, including protecting gastric mucosa [9], reducing liver fatty lesions [10], preventing oxidative stress [11], inhibiting the growth of colon cancer cells [12], anti-radiation [13], anti-adsorption of heavy metals [14]. It was previously reported that oral administration of AAM had an obvious hepatoprotective effect to alleviate alcoholic liver injury in mice [15]. However, as a kind of biological macromolecule, AAM is likely to play a role in protecting the liver by regulating the intestinal microbiome. Until now, the protective effects and possible mechanism of oral AAM on the intestinal flora and liver metabolome of mice with alcoholic liver injury are still unclear.

Intestinal microbiota is closely related to the immune, nerve and endocrine cells, and forms a highly complex intestinal ecosystem to maintain a state of balance. The normal intestinal micro-environment can prevent the adhesion and invasion of pathogenic bacteria [16]. Accumulating evidence demonstrates that excessive alcohol intake may destroy the intestinal microecological stability [17], and influence the metabolic function of intestinal microorganisms, such as short-chain fatty acids and bile acids [18]. Previous study has shown that the abundance of *Faecalibacterium prausnitzii* in the intestines of alcoholics was lower, while the abundance of *Enterobacteriaceae* was higher [19]. Besides, long-term alcohol intake may significantly change the composition of intestinal flora and its metabolites such as short chain fatty acids (SCFAs) and bile acids (BAs) in human [20]. Previous study has proved that intestinal ecological imbalance is closely related to the development of liver disease [21]. The intestinal microbiota has been reported to be one of the key influencing factors in the formation of ALD. Previous study proved that *Bifidobacterium* and *Akkermansia* can improve the function of intestinal barrier and protect the liver of mice with excessive alcohol intake [22,23]. Therefore, targeting intestinal microbiota with dietary intervention would be one of the effective approaches for the prevention or treatment of ALD. In vitro and in vivo study had confirmed that AAM has ameliorative effects on the pathological development of acute alcoholic liver injury [24]. However, few systematic study tried to describe the beneficial effect of AAM on the intestinal microbial composition and the association of intestinal microbiota with the hepatoprotective effects. 

The aim of this study is to investigate the protective effect and mechanism of AAM against alcoholic liver injury by high-throughput sequencing and liver metabolomics based on quadrupole time of flight mass spectrometer (UPLC-QTOF/MS). The connections between intestinal microbial phylotypes and lipid metabolic parameters were also revealed through correlation network, providing a strong theoretical basis for developing functional foods to improve alcoholic liver injury.

## 2. Materials and Methods

### 2.1. Preparation of AAM

*A. auricula* melanin (AAM) was prepared according to the previous study with minor modification [15]. Briefly, the powders of *A. auricula* fruiting body were poured into NaOH solution (1.25 mol/L) at material to liquid ratio of 1:35. The supernatant was obtained after ultrasonic treatment (200 W) for 60 min and centrifugation (2400× *g* at 4 °C) for 10 min. The pH of supernatant was then adjusted to 1.5 with HCl solution (2.0 mol/L), and then put into a water bath at constant temperature of 80 °C for 10 h to remove residual carbohydrates and proteins. After centrifugation at 4 °C for 10 min, the precipitate was removed and the supernatant was stored in the refrigerator (−20 °C). Melanin was further purified by extraction with a series of organic solvents and ultra-pure water according to the previously reported method [15]. The obtained precipitate was re-dissolved with NaOH solution (0.1 mol/L), and then adjusted to pH 7.0 with HCl solution (0.1 mol/L) under ultrasonic treatment. After dialysis, the solution was freeze-dried to obtain purified AAM for structure characterization and animal experiment.

### 2.2. Structural Characterization of AAM by Ultraviolet-Visible, Fourier Transform Infrared (FT-IR), Gel Permeation Chromatography (GPC) and Nuclear Magnetic Resonance (NMR) Spectroscopy

After the preparation of AAM solution (5 mg/mL), the absorption property of AAM was analyzed with ultraviolet-visible spectrophotometry, and the recording wavelength range was 200–1100 nm. Characteristic absorption of AAM powder was also carried out by FT-IR in the range of 4000–400 cm^−1^ at the revolution of 2 cm^−1^. The molecular weight of *Auricularia auricula* melanin was measured by GPC with ultrahydrogel linear columns (300 mm × 7.8 mm) and Waters 2414 refractive index detector. The 1H NMR spectra of AAM was recorded on a Bruker AVANCE NEO 600 using tetramethylsilane as an internal standard.

### 2.3. Animal Experimental Design

Specific pathogen-free Kunming mice (male, 6 weeks, weight 20 ± 2 g) were provided by Fujian Wushi Laboratory Animal Co., Ltd. (Fuzhou, China). All mice were reared in a SPF grade animal laboratory. The animal laboratory temperature was maintained at approximately 22 °C with a 12 h light/dark cycle. All mice werefree to get normal diet and water. After one week’s acclimation, all mice were randomly divided into four groups (*n* = 10 for each group), including control group, model group, low-dose AAM group (AAM-L, 50 mg/kg body weight), and high-dose AAM group (AAM-H, 150 mg/kg body weight). Mice of the model, AAM-L and AAM-H groups were orally received with 50% ethanol (5 mL/kg body weight) alone or in combination with administration of AAM every day, while mice of the control group were only received equal amount of normal saline. The mice were reared for 6 weeks, fresh feces samples were collected and placed into an ultra-cold storage freezer (−80 °C). After diet intervention for 6 weeks, all mice were fasted for 12 h and euthanized under anesthesia. Fresh blood, liver and cecal contents were collected from each mouse and put into 2.0 mL sterile centrifuge tubes. After resting the blood for 0.5 h at room temperature, centrifugation was performed and the serum was taken for preservation. Liver samples were weighed right away, quick-freezed by liquid nitrogen and finally maintained at −80 °C until further analysis. 

### 2.4. Biochemical Assays of the Serum and Liver Samples

Serum concentrations of triglyceride (TG), total cholesterol (TC), low density lipoprotein cholesterol (LDL-C), high density lipoprotein cholesterol (HDL-C), aspartate aminotransferase (AST), and alanine aminotransferase (ALT) were tested with an automatic biochemical analyzer following the operating instructions provided by the commercial kits of Nanjing Jiancheng Bio. Ins. (Nanjing, China). A small piece of liver sample (0.1 g) was put into physiological saline (0.9 mL) and homogenized by high-speed homogenizer to obtain liver homogenate. The supernatants were collected by centrifugation at 1000× *g* and 4 °C for 10 min. Liver TG, TC, glutathione (GSH), superoxide dismutase (SOD), lactic dehydrogenase (LDH), malondialdehyde (MDA) and bile acid (BAs) were then determined through corresponding commercial kits provided by Nanjing Jiancheng Bio. Ins. (Nanjing, China).

### 2.5. Hematoxylin–Eosin (H&E) Staining of Liver and Ileum Section

For histopathological evaluation, liver and ileum samples were fixed with 4% paraformaldehyde solution for 24 h, and then embedded in paraffin and cut into section with 4 μm thickness through a series of ethanol and xylene solutions. After the fabricated sections were stained with H&E, they were observed and photographed under light microscope (Nikon, Tokyo, Japan).

### 2.6. High Throughput Sequencing Analysis of Intestinal Flora

Fecal genomic DNA extraction kit (MoBio, Carlsbad, CA, USA) was used to extract DNA from samples of cecal contents. The V3-V4 hypervariable regions of 16S rDNA were sequenced by high throughput sequencing based on Illumina MiSeq platform at Shanghai Majorbio Co., Ltd. (Shanghai, China). The sequencing raw data were imported into QIIME 2 software, and the filtered sequences were clustered into operation taxon units (OTU) with 97% identity threshold. Based on the Greengenes database (version 13.8), the sequence similarity was matched to identify microbial phylotypes at the genus level, and the relative abundance of each OTU was obtained. The relative abundance of different OTUs among different groups were revealed at the genus level using STAMP software (Ver. 2.1.3). R software (Ver.3.5.1) and Cytoscope software (ver.3.6.0) were used to perform the correlation analysis between the key intestinal microbial phylotypes and lipid metabolism parameters.

### 2.7. Determination of Short Chain Fatty Acids (SCFAs)

Fecal SCFAs were extracted and detected using our previously reported method with appropriate modifications [25]. Briefly, 500 μL saturated NaCl solution was added to 0.05 g feces and placed at room temperature for 0.5 h, followed by homogenization on a high-speed homogenizer for 3 min. Then 20 μL H_2_SO_4_ (10%, *v*/*v*) was added and mixed by vortexing for 30 s. The total SCFAs were completely collected with 800 μL anhydrous ether and then centrifuged (10,000× *g*, 15 min, 4 °C). Finally, the residual water in the supernatant was removed with anhydrous Na_2_SO_4_, and the contents of SCFAs in the supernatant were determined by gas chromatography.

### 2.8. Metabolomics Analysis of Liver

Liver sections (25 mg) were extracted at the mixed organic solvent (acetonitrile: methanol: water = 2: 2: 1), and vibrated on a vortex oscillator for 1 min. After centrifugation at 10,000× *g* for 15 min at 4 °C, the supernatant was taken into a centrifuge tube and evaporated to dryness at 37 °C under a gentle stream of nitrogen. The dried samples were reconstituted in 200 μL of 50% acetonitrile, and centrifugated at 12,000× *g* for 10 min at 4 °C. The supernatant was injected into the UPLC-QTOF/MS system for analysis. UPLC separation was carried out on Agilent 1290 Infinity series UPLC System equipped with a UPLC BEH Amide column (2.1 × 100 mm, 1.7 μm, Waters) and a 6530 QTOF electrospray ionization MS system (Agilent, Santa Clara, CA, USA).

Raw data obtained from UPLC-QTOF/MS system were converted and analyzed by MPP software (Agilent, CA, USA) for peak detection, alignment and identification. The peak intensities were exported to SIMCA 15.0 software for multivariate statistical analysis including PCA, PLS-DA and OPLS-DA. Liver metabolites of significant difference between the model and AAM-L groups were selected by OPLS-DA and volcano plot (VIP value > 1.0, *p* < 0.05 and fold change > 2). The selected metabolites were then identified with HMDB database (http://hmdb.ca). Pathway analysis of liver metabolites with significant difference was performed on MetaboAnalyst 5.0 (http://www.metaboanalyst.ca).

### 2.9. Reverse Transcription-Quantitative Polymerase Chain Reaction (RT-qPCR)

Total RNA was extracted from liver by Trizol reagent from Takara Biomedical Technology Co., Ltd. (Beijing, China) and quantified using NanoDrop Spectrophotometer (Thermofisher, Waltham, MA, USA). cDNA was used in PrimeScriptTM RT reagent Kit (Takara Bio company, Kusatsu, Japan). The mRNA levels of cluster of differentiation 36 (*CD36*), interferon-γ (*IFN-γ*), acetyl-CoA carboxylase 1 (*ACC1*), carnitine palmitoyltransferase-1 (*CPT-1*), fatty acid synthase (*FASn*), low density lipoprotein receptor (*LDLr*) and tumor necrosis factor-α (*TNF-α*) were detected in the StepOne Real-time PCR system (Applied Biosystems, Waltham, MA, USA). The PCR conditions were as following: initial activation 95 °C for 30 s, denaturation 95 °C for 5 s, annealing 55 °C for 15 s, extension 72 °C for 15 s, 40 cycles. Mouse 18S rRNA gene was used as an internal control for sample normalization. The 2−ΔΔCT standard method was used for analyzing the real-time qPCR data. The primer sequences of real-time qPCR were shown in Table 1.

### 2.10. Statistical 

All data of this study are indicated as the mean ± standard deviation. Statistical significance of the results were analyzed by SPSS software (ver. 22.0, IBM Corp, Somers, NY, USA) using the one-way analysis of variance (ANOVA). The significance level in the analyses was considered *p* < 0.05 (significant) and *p* < 0.01 (extremely significant).

## 3. Results and Discussion

### 3.1. Morphological Analysis of AAM

#### 3.1.1. UV-Visible Absorption Spectra of AAM

The optical density of AAM in ultraviolet region gradually decreased with the increase of wavelength (Figure 1A). There was an small acromion near the 275 nm, which may resulted from the absorption of the aromatic amino acid residues (tyrosine, tryptophan, phenylalanine) in the binding protein of melanin. Generally speaking, the most obvious characteristic of melanin is the strong absorption over a broad wavelength range by high conjugate regions. Therefore, the linear slope of wavelength-absorbance can be used to preliminarily identify the purified substance as melanin. The logarithm of the optical density of AAM solution produced a highly linear curve with the wavelength, and its negative slope was −0.002036 (Figure 1B), which falls within the range previously reported in the literature (−0.0015 to −0.0030) [26]. 

#### 3.1.2. FT-IR Spectrometric Analysis and Molecular Weight Determination of AAM

FT-IR spectrum showed that the characteristic absorption peaks of AAM were 3419.88 cm^−1^ and 1647.88 cm^−1^, indicating the presence of -OH and -NH2 groups (Figure 1C). The small peaks at 2925.78 cm^−1^ and 2854.18 cm^−1^ may be resulted from the oscillation of CH_2_ and CH_3_ groups. The absorption at 1647.88 cm^−1^ may be due to the tensile vibration of aromatic C=C bond and symmetrical tensile vibration of -COO-. The absorption at 1455.53 cm^−1^ was caused by the deformation of aliphatic C-H, which was caused by bending vibration of C-H bond on benzene ring; 900–700 cm^−1^. The vibration absorption peak of hydrogen on fatty carbon was below 700 cm^−1^. In the process of acid precipitation, some narrow and sharp peaks may appear due to the destruction of some debris. The average molecular weight of AAM was measured by gel permeation chromatography (GPC) and calculated to be 49.03 kDa based on polyethylene glycol with different molecular weight (Figure 1D).

##### H-NMR Spectrum and Elemental Analysis of AAM

The 1H-NMR spectrum of AAM (Figure 1E) displayed the signals of aliphatic and aromatic regions. According to 1H NMR spectrum, the peaks at δH 0.5 ppm to δH 2.5 ppm were attributed to CH_3_ and CH_2_ groups. Peaks at δH 3.5–4.5 ppm can be attributed to protons on carbons attached to oxygen or nitrogen atoms. The solvent peak of D_2_O is at δH 4.70 ppm. The peaks at δH 7.13 ppm and δH 6.88 ppm were attributed to the signals of melanin polymeric chain for protons of indole or pyrrole structural units. In conclusion, the 1H-NMR spectrum of AAM in this study is agreement with that of natural black fungus melanin previously reported by others [27]. Besides, result of elemental analysis suggested that AAM possesses C, H, N and O elements, and almost no S element, which indicates that there are a lot of eumelanin in AAM (Table 2).

### 3.2. Effects of AAM on Body Weight Growth and Liver Index

The body weight and liver index of mice of the control and experimental groups are shown in Figure 2A. Three weeks after intragastric administration of alcohol (50%, *v*/*v*), the weight growth rate was clearly lower than that of the control group. After 6 weeks of intervention, there was a significant difference in the body weight gain between the model group and the control group (*p* < 0.01). Alcohol intake also induced an obvious increase of liver index compared to the control group (*p* < 0.01), while oral administration of AAM obviously reduced the liver index of mice exposed to alcohol intake (*p* < 0.01).

### 3.3. Effects of AAM on Serum Biochemical Profile

As illustrated in Figure 2B, the serum concentrations of TC, LDL-C, AST and ALT were obviously elevated in the model group as compared with the control group, while the serum concentrations of HDL-C were significantly decreased after 6 weeks of alcohol intervention. The serum levels of TG, TC and LDL-C in mice of the AAM-L and AAM-H groups were significantly lower than those in the model group, and closer to those mice in control group, indicating that AAM could effectively prevent the lipid metabolism disturbance caused by excessive alcohol intake. Furthermore, serum ALT and AST concentrations are the most common indicators for evaluating the degree of liver injury. Alcohol stress leads to slight necrosis and autolysis of hepatocytes, and a large number of transaminases (such as ALT and AST) will be released, resulting in a sharp increase in transaminase activity outside the liver cells and in serum. Therefore, the increase in serum transaminase activity can reflect the degree of hepatocyte damage [28]. In this study, the serum concentrations of ALT and AST were significantly reduced in the AAM-L and AAM-H groups as compared with that the model group (*p* < 0.05), which is closer to the control group. Therefore, AAM intervention obviously inhibited the necrosis of hepatocytes caused by alcohol intake, and weaken the degree of hepatocyte damage. Besides, the serum concentrations of HDL-C in mice of the AAM-L and AAM-H groups were significantly higher than those of the model group (*p* < 0.01).

### 3.4. Effects of AAM on Liver Biochemical Parameters

As illustrated in Figure 3, compared with the control group, the concentrations of hepatic TG, TC and LDH were significantly elevated in mice of the model group, whereas the hepatic concentrations of SOD and GSH were significantly decreased. However, the concentrations of hepatic TG and TC in the liver of mice treated with AAM were close to those in the control group, but lower than those in the model group, suggesting that AAM could alleviate the abnormal increase of TG and TC in the liver caused by alcohol intake to a certain extent. AAM could significantly improve the pathological changes of liver tissue. Abnormal increase of liver LDH activity is mainly seen in myocardial infarction, acute and hepatitis, liver cancer (especially metastatic liver cancer), skeletal muscle disease, blood system disease, pulmonary infarction, hypothyroidism, nephrotic syndrome and advanced malignant tumor [29]. LDH activity was often increased when hepatocytes were injured. Our experimental data suggested that the LDH activity in mice of the model group was obviously higher than that of the other groups. The LDH level of the liver in mice of the AAM-L and AAM-H groups was close to that of the control group, implying that oral administration of AAM could effectively inhibit the liver damage caused by alcohol intake. 

Lipid peroxidation mediated by alcohol exposure is an important cause of liver injury. MDA is a by-product of lipid peroxidation and its level in liver can indirectly reflect the degree of liver injury. In this study, the hepatic MDA level in the model group was evidently higher than the control, AAM-L and AAM-H groups (*p* < 0.01), and the hepatic MDA levels of the AAM-L and AAM-H groups were close to that of the control group, indicating that alcohol exposure can cause liver lipid peroxidation (Figure 3F). AAM had a significant inhibitory effect on MDA production induced by alcohol intake. There is an antioxidant defense system composed of SOD and GSH in hepatocytes, which has a strong capability to scavenge free radicals produced in the body. When the content of free radicals exceeds its own maximum scavenging capacity, it will lead to the accumulation of free radicals, thus leading to oxidative damage [30]. In this study, excessive alcohol exposure significantly inhibited the activity of hepatic SOD compared with the control group. Oral administration of AAM evidently elevated the hepatic activities of SOD and GSH, thus achieving the protective effect on the liver. In addition, bile acids (BAs) play a crucial role in the process of alcohol-related liver disease (ALD). Alcoholic liver injury is usually accompanied by cholestasis and high levels of BAs in the liver. Our results exhibited that the liver BAs in the model group were significantly deposited, and the total BAs content in the liver of the mice was significantly decreased after AAM intervention (Figure 3G).

### 3.5. Effects of AAM on Histopathological Features of Liver and Small Intestine

The above results indicate that oral administration of low-dose AAM (50 mg/kg per day) can effectively prevent alcoholic liver injury, which was comparable to that of high-dose AAM (150 mg/kg per day), so in the next experiment we only focused on mice exposed to alcohol intake and supplemented with low-dose AAM. 

As shown in Figure 4A, the pathological sections of liver exhibited that the hepatocytes of normal mice in the control group were arranged orderly with clear outlines, no deformation and necrosis, large and round nuclei, and clear liver lobules. By comparison, in the model group, the liver cords were arranged disorderly, the liver cells were turbid and swollen, and the cytoplasm was loose and balloon-like, indicating that alcohol intake may lead to liver damage. The liver tissue morphology of mice supplemented with low-dose AAM for 6 weeks was close to that of the control group, the degree of cell necrosis was reduced, and the liver cords were arrayed in an orderly manner, radiating from the central vein to the surrounding. As shown in Figure 4B, the histopathological characteristics of the small intestine by H&E staining showed that no pathological features were observed in the small intestine in mice of the control group. The villus length of the small intestine of the model group was obviously shorter than that of the control group, while the villus length of the AAM-L group was closer to that of the control group. The integrity of the small intestinal epithelial cells in mice with alcohol treatment was obviously damaged by alcohol intake, the number of epithelial cells was reduced, the villus length was shortened, and the basal crypts were atrophied. Low-dose AAM significantly improved the histopathological changes in the small intestine caused by alcohol intake.

### 3.6. AAM Modulated Intestinal Microbiota in Mice Exposed to Alcohol Intake

The relative proportions of 16 key microbial phylotypes were obviously altered in mice of the model and control groups, including 6 obviously increased taxon and 10 obviously reduced taxon (Figure 5A). As compared with the control group, mice of the model group were characterized by higher proportion of *Dubosiella*, *Muribaculum*, *Flavobacteriaceae*, *Escherichia-Shigella*, *Gordonibacter* and *Ruminiclostridium*_1, but lower amount of *Alloprevotella*, *Roseburia*, *Prevotellaceae*_UCG-001, *Tyzzerella*, *Peptococcus*, *Negativibacillus*, *Staphylococcus*, *Bilophila*, *Jeotgalicoccus* and *Hydrogenoanaerobacterium*, implying that intestinal microbial disorder occurred in mice exposed to alcohol intake. Low-dose of AAM treatment (50 mg/kg per day) significantly changed the relative abundance of 20 key microbial phylotypes, including 15 up-regulated taxon and 5 down-regulated taxon (Figure 5B). Interestingly, supplementation of AAM (50 mg/kg per day) obviously elevated the relative proportions of *Lachnospiraceae*_NK4A136_group, *Turicibacter*, *Lachnospiraceae*_UCG-006, *Ruminococcaceae*_UCG-010, *Bifidobacterium*, *Akkermansia*, *Ruminococcaceae*_UCG-004, *GCA-900066225*, *Bilophila*, *Romboutsia*, *Family_XIII*, *Muribaculaceae*, *Erysipelotrichaceae*, *Rodentibacter*, *Hydrogenoanaerobacterium*, but dramatically decreased the relative levels of *Lactobacillus*, *Lachnospiraceae*_UCG-001 and *Eggerthellaceae* in mice with excessive consumption of alcohol. Accumulating evidence demonstrated that alcohol-related liver disease is highly associated with intestinal microbial disorder [31]. Evidence suggests that the development of ALD can be reduced by controlling the abundance of some harmful bacteria in the intestinal flora [32]. Excessive intake of alcohol can cause intestinal ecological imbalance, destroy the permeability of intestinal barrier, which leads to the transfer of microorganisms and their metabolites to the liver through the portal vein and promotes liver injury [33]. 

The proportion of *Lactobacillus* in the intestinal flora of model group was obviously higher than that of AAM-L group (Figure 5B). Although *Lactobacillus* is a common probiotics, an significant increase of *Lactobacillus* is also known to be highly associated with hepatic steatosis [34]. Even the increase of *Lactobacillus* is positively correlated with liver cirrhosis in alcohol dependent patients [34]. In addition, after AAM administration, the abundance of several common probiotics such as *Akkermansia* and *Bifidobacterium* were increased significantly. *Akkermansia* is a kind of beneficial bacteria that can increase the thickness of mucus and enhance the barrier function of the intestinal tract [35]. *Akkermansia* deficiency has been identified as an early marker of alcohol-induced intestinal dysbacteriosis [36]. It has been proved that *Akkermansia* can improve the intestinal barrier and protect against ethanol-induced liver injury [37]. *Bifidobacterium* can suppress the reproduction of harmful bacteria, ameliorate gastrointestinal barrier function and reduce the secretion of proinflammatory cytokines [38]. In addition, the abundance of *Romboutsia* and *Muribaculaceae* were increased significantly in mice of the AAM-L group. Of which, *Romboutsia* is a common bacteria in the gut and highly related to body energy metabolism [39]. The abundance of *Bilophila* and *Hydrogenoanaerobacterium* in the control group and AAM-L group were significantly higher compared with the model group. It has been reported that *Bilophila* was related to the liver fat content in mice [40]. Interestingly, oral AAM supplementation significantly increased the relative proportions of SCFAs producers, including *Lachnospiraceae*_NK4A136, *Ruminococcaceae*_UCG-010 and *Ruminococcaceae*_UCG-004 in mice with excessive alcohol consumption. SCFAs are the main energy source of intestinal epithelial cells and participate in many physiological properties, such as mucosal protection, immune regulation and metabolism [41]. 

Moreover, the fecal levels of butyric acid, isobutyric acid and n-valeric acid in mice were elevated significantly after AAM supplementation for 6 weeks (Figure 6). Since the production of SCFAs can be utilized in colonocytes for sustenance, *Lachnospiraceae*_NK4A136 was considered as anti-inflammatory factors [42]. Notably, the genus *Ruminococcaceae*_UCG-010 and *Ruminococcaceae*_UCG-004 belong to the family *Ruminococcaceae* can also produce high levels of SCFAs and reduce inflammation [43]. *Ruminocococeae* is highly correlated with intestinal health and is abundant in healthy people [44]. These results suggest that oral AAM supplementation can protect the intestinal mucosal barrier, reduce the entry of alcohol and intestinal harmful metabolites into the blood through the intestinal barrier by regulating the homeostasis of intestinal flora and increasing the relative abundance of SCFAs producers.

### 3.7. Effects of AAM on the mRNA Levels of Lipid Metabolism and Inflammatory Response Related Genes in Liver

Results of RT-qPCR of lipid metabolism and inflammatory response related genes in liver showed that alcohol intake promoted the mRNA levels of *ACC1*, *FASn*, *CD36*, *IFN-γ* and *TNF-α* in liver, while inhibited the mRNA levels of *CPT-1* and *LDLr* in liver. Conversely, low dose AAM intervention can reverse the expression of these genes and improve the degree of liver injury caused by alcohol intake (Figure 7). *ACC1* and *FASn* are two critical enzymes closely associated with lipogenesis, helping acetyl-CoA to form palmitic acid desaturated by SCD1 [45]. *CPT1* is a critical rate-limiting enzyme for controlling the rates of fatty acid oxidation [46]. In addition, the decreased *ACC1* level may promote the transcription of *CPT-1* in liver, and then regulate liver lipid metabolism, especially the accumulation of fatty acids. It has also been reported that fatty acid accumulation and metabolic imbalance is one of the pathogenesis of ALD, and an increase in the production of fatty acids can cause excessive synthesis of TG in liver [47]. *FASn* is mainly responsible for catalyzing the synthesis of long-chain fatty acids, which is less expressed in normal cells, but highly expressed in some pathological conditions such as fatty liver [48]. As a multifunctional membrane protein, fatty acid translocase *CD36* promotes the absorption of long-chain fatty acids (LCFA) and plays a key role in controlling the rate of lipid metabolism [49]. Therefore, inhibition of *CD36* expression contributes to the improvement of lipid accumulation in human hepatocytes [50]. Moreover, as a recognized receptor for fatty acid absorption and scavenger receptor for oxidized lipids, *CD36* is known to promote triglyceride accumulation and subsequent lipid induced endoplasmic reticulum stress [51]. *IFN-γ*, one of the most effective cytokines to strongly induce liver injury [52], is a pro-inflammatory cytokine that can enhance the sensitivity of hepatocytes to *TNF-α*. Therefore, high *IFN-γ* level may cause damage to liver regeneration and make hepatocytes vulnerable to damage [53]. Liver plays a critical role in maintaining the cholesterol homeostasis, and excessive alcohol intake has side effects on cholesterol metabolism. It is well known that the liver removes excess low-density lipoprotein from the blood circulation through the regulation mechanism of low-density lipoprotein receptor (LDLr) [54]. It is reported that LDL-LDLr interaction help to eliminating the circulating LDL-C, and inhibiting the expression of LDLr in liver can elevate the risk of familial hypercholesterolemia [55]. The elevation of cholesterol concentrations in hepatocytes palys a critical role in the occurrence and process of alcoholic liver disease. As the main metabolite of ethanol metabolism in the liver, acetaldehyde can elevate the production of intracellular cholesterol by inducing endoplasmic reticulum stress, making hepatocytes sensitive to *TNF-α* and inducing cytotoxicity.

### 3.8. Correlations of Microbial Phylotypes with Metabolic Parameters and Liver Gene Transcription Profile

The correlation analysis based on heatmap and network indicated significant correlations between the key microbial phylotypes with metabolic parameters and liver genes transcription (Figure 8). Interestingly, *Muribaculaceae*, *Lachnospiraceae*_UCG-006, *Ruminococcaceae*_UCG-004, *Rodentibacter*, *Lachnospiraceae*_NK4A136, *Akkermansia*, *Turicibacter* and *Romboutsia* were positively correlated with the mRNA level of *LDLr* and negatively related to serum LDL-C levels. *LDLr* gene is involved in the homeostasis of LDL-C level in plasma. Down-regulation of *LDLr* gene expression will reduce the ability of liver to clear circulating LDL-C, thus aggravating the degree of liver injury. In addition, *Akkermansia* was positively correlated with the liver SOD activity and serum HDL-C level, but negatively correlated with liver TG level, hepatic mRNA levels of *CD36* and *IFN-γ*. Besides, *Bilophila* and *Lachnospiraceae*_NK4A136 were positively correlated with the liver SOD and serum HDL-C, and *Lachnospiraceae*_NK4A136 was also negatively correlated with liver TG level.

### 3.9. Effects of AAM on the Liver Metabolomic Profiling in Model Mice

In order to visualize the differences of metabonomics among different experimental groups, the metabonomics data (ESI+ & ESI-) of liver samples were analyzed by multivariate statistics analysis including principal component analysis (PCA), partial least squares discrimination analysis (PLS-DA) and orthogonal PLS-DA (OPLS-DA) (Figure 9; Figure 10). PCA and PLS-DA score plots demonstrated significant differences in liver metabolic spectrum between different experimental groups in positive-ion and negative-ion modes (Figure 9A,B and Figure 10A,B). As shown in Figure 9C and Figure 10C, the OPLS-DA score plots showed that the AAM-L group were clearly separated from the model group, indicating that AAM administration can improve the liver metabolic disorder in mice exposed to alcohol intake. The S-plots of OPLS-DA showed the difference variables between the model and AAM-L groups (VIP > 1.0 and *p* < 0.05) (Figure 9D and Figure 10D). Volcano plots demonstrated significant difference in liver metabolites between the model and AAM-L groups (fold change > 2 and *p* < 0.05) (Figure 9E and Figure 10E). In the positive-ion mode, 39 potential biomarkers in liver were successfully identified between the model and AAM-L groups (VIP value > 1.0, *p* < 0.05 and fold change > 2). Compared with the model group, 13 liver metabolites were significantly decreased and 26 metabolites were significantly increased in mice of the AAM-L group (Figure 9F). In the negative-ion mode, 28 potential biomarkers were found between the model and AAM-L groups, of which 18 were increased significantly and 10 were reduced significantly in mice of the AAM-L group (Figure 10F). In order to make a more intuitive view of the metabolic pathway affected by AAM treatment in mice with excessive alcohol consumption, liver metabolites that differ significantly between groups were enriched by metabolomics data analysis platform-MetaboAnalyst (Figure 9G and Figure 10G). In ESI+, the metabolic pathways significantly altered by AAM treatment mainly included fatty acid degradation, phosphatidylinositol signaling system, ascorbate and aldarate metabolism, starch and sucrose metabolism, galactose metabolism, amino sugar and nucleotide sugar metabolism, alpha-linolenic acid metabolism, glycolysis/gluconeogenesis, fructose and mannose metabolism, biosynthesis of unsaturated fatty acids, purine metabolism, inositol phosphate metabolism and phenylalanine metabolism compared with the model group. In ESI-, purine metabolism, alpha-linolenic acid metabolism, pantothenate and CoA biosynthesis, glycerolipid metabolism, glycerophospholpid metabolism, steroid hormohe biosynthesis, pyrimidine metabolism and primary bile acid biosynthesis were the main metabolic pathways remarkably altered by AAM treatment when compared with the model group. In the ESI- mode, biosynthesis of unsaturated fatty acids, sphingolipid metabolism, amino sugar and nucleotide sugar metabolism, cysteine and methionine metabolism were the main metabolic pathways significantly altered by AAM treatment by comparison with the model group.

In the ESI+ mode, the hepatic levels of 2-hydroxyphenylacetic acid, N-glycolylneuraminic acid and maltotriose were significantly up-regulated in mice exposed to alcohol by comparison with the control group, while oral AAM significantly reduced the contents of these metabolites. 2-hydroxyphenylacetic acid was reported as an endogenous biomarker to detect the pathogenesis and susceptibility of ALD [56]. N-glycolylneuraminic acid is highly expressed in cancer tissues of patients with breast cancer, melanoma and colorectal cancer [57]. The content of maltotriose in healthy body is low, but abnormal glucose metabolism can lead to significant up-regulation of maltotriose in patients. In addition, after 6 weeks of AAM treatment, phenylpropionylglycine, taurodeoxycholic acid, cholecalciferol, arachidonoyl ethanolamide (AEA), eicosapentaenoic acid, sparfloxacin, oleoyl-CoA and linoloyl ethanolamide in liver were significantly up-regulated compared with the model group. As an acyl glycine, phenylpropionylglycine is also known as a metabolites of fatty acid metabolism [58], indicating that AAM may regulate the abnormal lipid metabolism in liver by promoting fatty acid metabolism. As one of the main components of bile acids in liver, taurodeoxycholic acid exhibited a protective role in combating oxidative and endoplasmic reticulum stress [59]. Cholecalciferol has potential beneficial and anti-inflammatory effects on lipid metabolism, and one of its biological characteristics is antioxidation, which can significantly reduce ROS [60]. In addition, cholecalciferol can reduce the formation of endogenous harmful free radicals by enhancing the body’s antioxidant defense system (including enhancing the activities of GSH and SOD) [61]. Previous study also reported that cholecalciferol can down regulate the inflammatory cytokine TNF-α, thereby inhibiting ROS production [62]. Arachidonoyl ethanolamide is an endogenous cannabinoid, which is associated with many physiological diseases, including obesity, metabolic syndrome, liver disease and inflammation [63]. Eicosapentaenoic acid has health-promoting and beneficial preventive effects on many diseases, for instance metabolic disturbance and inflammation associate diseases. The presence of eicosapentaenoic acid can remarkably reduce the concentrations of cholesterol and LDL-C [64].

In the ESI- mode, the contents of maltotriose and 2’-deoxy-d-ribose (2’-dRib) were significantly down-regulated in mice of the AAM group. Previous study had shown that the level of maltotriose can reflect the abnormality of glucose metabolism to a certain extent [65]. As a sugar with a high reducing capacity, 2’-dRib can induce the cell toxicity and apoptosis of pancreatic cancer β cell line, 2-oxidative stress and proteoglycation are the important mechanisms of the injury of islet β cells induced by 2’-dRib [66]. Besides, oleoyl-CoA, linoleoyl-CoA, sphinganine, acamprosate, kynurenic acid and amygdalin in liver were significantly down-regulated in mice with alcoholic liver injury, while AAM treatment could significantly reverse these adverse changes [67]. Of which, sphinganine is an effective biomarker for early detection of liver fibrosis, and the content of sphinganine will decrease during liver fibrosis [68]. As a modulator of N-methyl-D-aspartate receptors, acamprosate is approved by the US Food and Drug Administration (FDA) for the medical treatment of alcohol dependence [69]. According to a previous study, acamprosate may work by normalizing NMDA-mediated dysregulation of glutamate neurotransmission, which may help reduce the pain and cravings associated with alcohol withdrawal [70]. Kynurenic acid is a tryptophan metabolite formed along the kynurenine metabolic pathway and has anti-inflammatory and antioxidant activities [71]. It can alleviate the inhibition of colitis [72], reduce the production of TNF-α and have a significant protective effect on hepatocytes [73]. It has been reported that amygdalin has obvious anti-inflammatory activity by reducing the secretion of TNF-α [74], effectively alleviating hepatic steatosis caused by endoplasmic reticulum stress. The above metabolomic analysis results show that AAM intervention can obviously improve alcoholic liver injury in mice with excessive alcohol consumption by regulating lipid metabolism, oxidative stress and inflammatory related metabolites.

## 4. Conclusions

In this research, the potential effects of AAM on alcohol-induced liver injury and their probable mechanisms were investigated. We found that AAM supplementation significantly improved alcoholic liver injury and intestinal microbial disorder in mice exposed to alcohol intake. The potential protective mechanisms of AAM against the pathological process of ALD were elucidated through liver metabonomics and RT-qPCR. These findings preliminarily revealed that oral administration of AAM ameliorates alcoholic liver injury by modulating the composition of intestinal microbiota (increasing the relative abundance of *Akkermansia* and *Bifidobacterium*) and liver metabonomic profile, and regulating the mRNA levels of lipid metabolism and inflammatory response related genes in liver. Taken together, this study implied that AAM has potential beneficial effects on preventing alcoholic liver injury and is expected to become a new functional food ingredient. In further study, we need to clarify the protective mechanism of AAM supplementation on alcoholic liver injury and oxidative stress through mouse model of humanization of intestinal flora, so as to provide reference for future clinical application.

## Figures and Tables

**Figure 1 foods-10-02436-f001:**
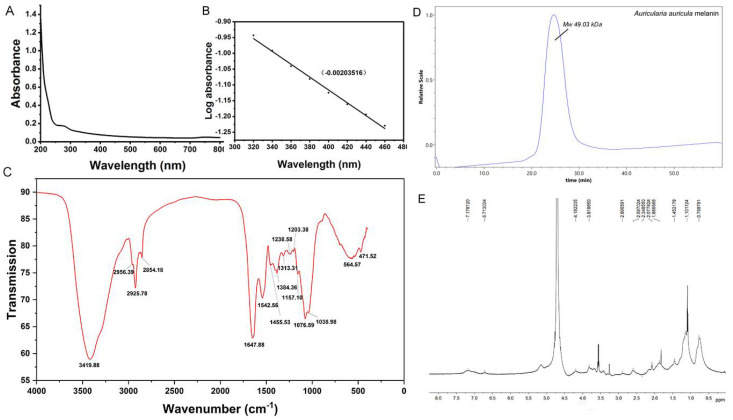
Structural characterization of *A. auricula* melanin by (**A**) ultraviolet-visible absorption spectrum, (**B**) Logarithm of absorbance to wavelength, (**C**) FT-IR spectrum, (**D**) Gel permeation chromatography (GPC) and (**E**) 1H NMR spectrum.

**Figure 2 foods-10-02436-f002:**
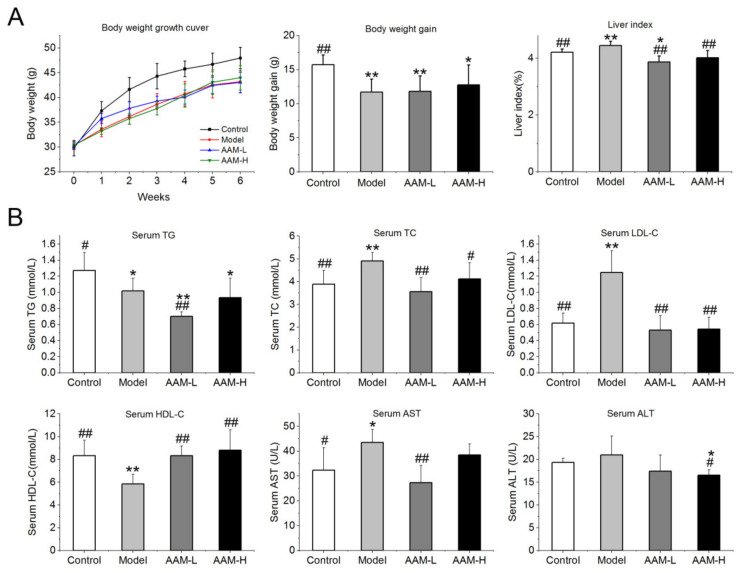
Effects of AAM administration on physiological parameters and the serum biochemical profile in mice exposed to alcohol intake for consecutive 6 weeks. (**A**) Bodyweight growth curve, body weight gain and liver index of mice; (**B**) Serum levels of TG, TC, LDL-C, HDL-C, AST and ALT. ## *p* < 0.01 and # *p* < 0.05, versus the Model group; ** *p* < 0.01 and * *p* < 0.05, versus the Control group.

**Figure 3 foods-10-02436-f003:**
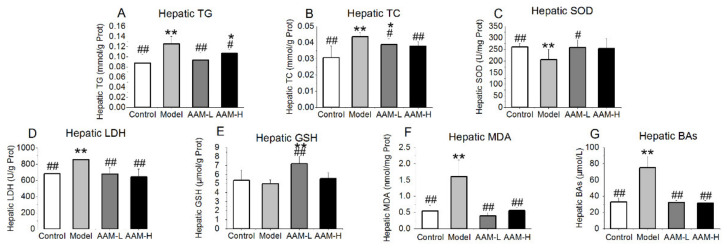
Effects of AAM administration on the liver biochemical profile in mice exposed to alcohol intake for consecutive 6 weeks. (**A**) TG; (**B**) TC; (**C**) SOD; (**D**) LDH; (**E**) GSH; (**F**) MDA; (**G**) BAs. ^##^
*p* < 0.01 and ^#^
*p* < 0.05, versus the Model group; ** *p* < 0.01 and * *p* < 0.05, versus the Control group.

**Figure 4 foods-10-02436-f004:**
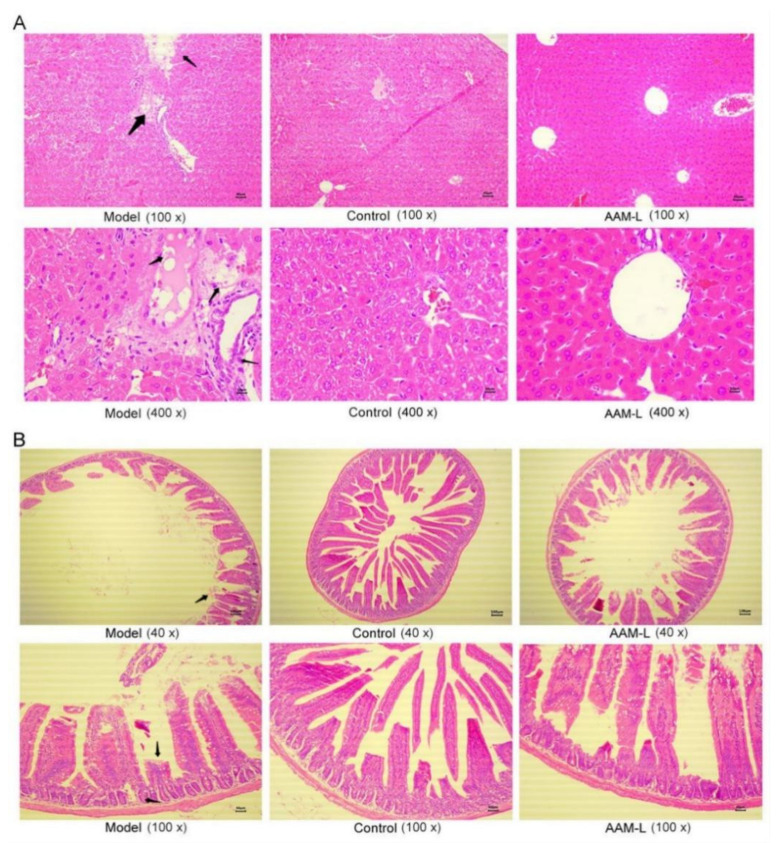
Effects of AAM administration on histopathological features of liver and small intestine in mice exposed to alcohol intake for consecutive 6 weeks. (**A**) Histopathological features of liver; (**B**) Histopathological features of small intestine.

**Figure 5 foods-10-02436-f005:**
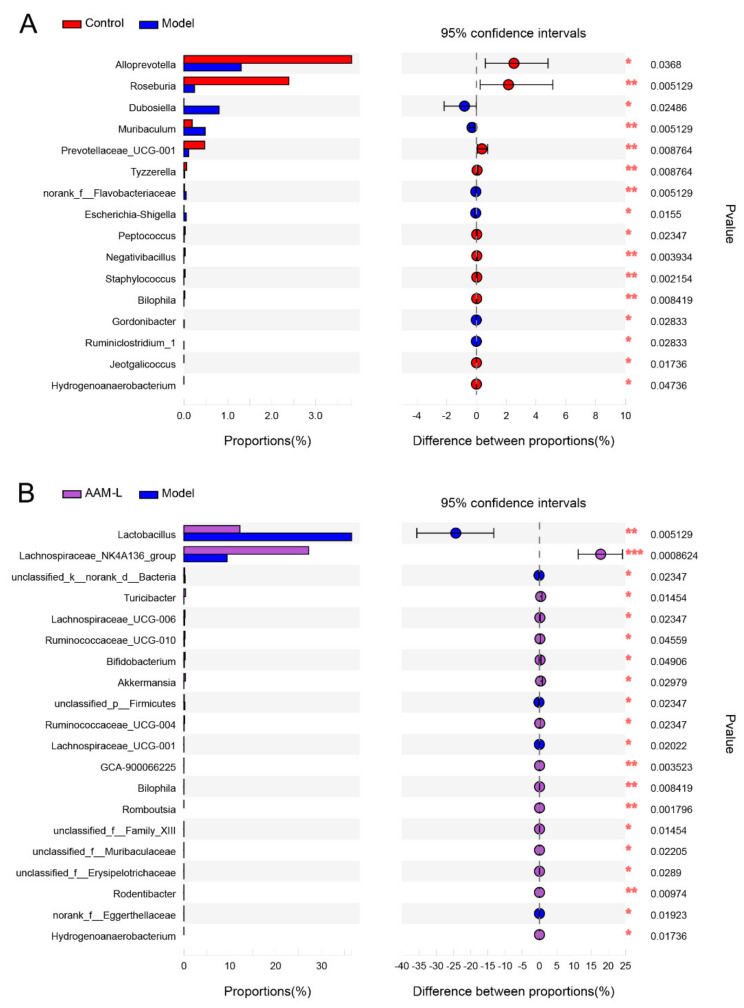
Effect of AAM administration on intestinal microbial population in mice exposed to alcohol intake for consecutive 6 weeks. (**A**) The Control group versus the Model group; (**B**) The AAM-L group versus the Model group; *** *p* < 0.001, ** *p* < 0.01 and * *p* < 0.05.

**Figure 6 foods-10-02436-f006:**
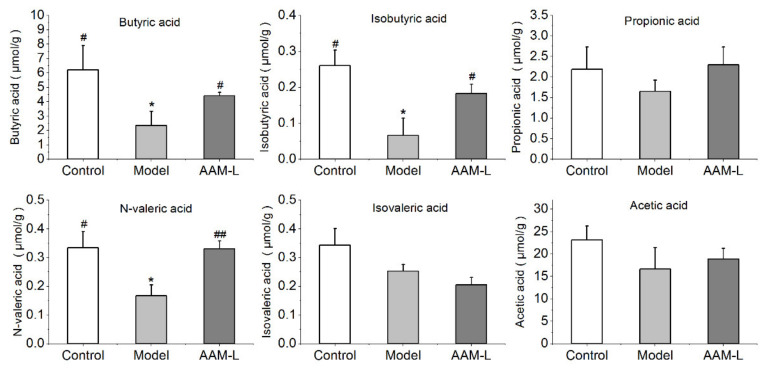
Effects of AAM administration on the content of intestinal short-chain fatty acids (including acetic acid, propionic acid, butyric acid, isobutyric acid, n-valeric acid and isovaleric acid) in mice exposed to alcohol intake for consecutive 6 weeks. ^##^
*p* < 0.01 and ^#^
*p* < 0.05, versus the Model group; * *p* < 0.05, versus the Control group.

**Figure 7 foods-10-02436-f007:**
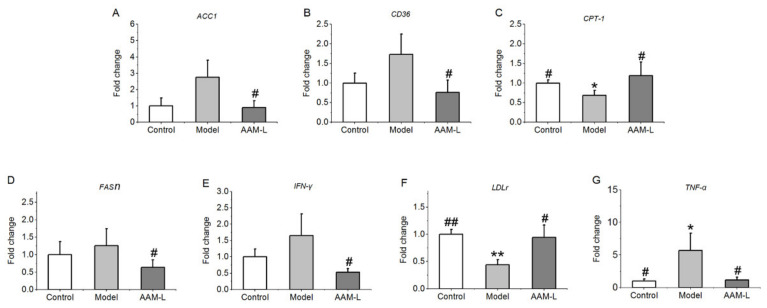
Effects of AAM administration on mRNA levels of hepatic genes in mice with ALD for consecutive 6 weeks. (**A**) *ACC1*; (**B**) *CD36*; (**C**) *CPT-1*; (**D**) *FASn*; (**E**) *IFN-γ*; (**F**) *LDLr*; (**G**) *TNF-α*. ## *p* < 0.01 and # *p* < 0.05, versus the Model group; ** *p* < 0.01 and * *p* < 0.05, versus the Control group.

**Figure 8 foods-10-02436-f008:**
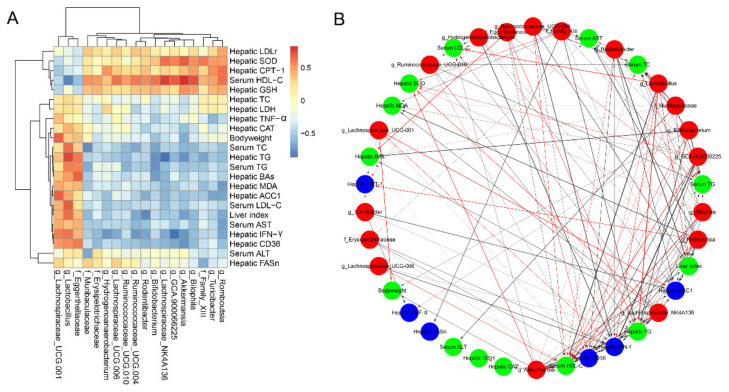
Spearman’s correlation of the key intestinal microbial phylotypes with metabolic parameters and liver gene transcription profile. (**A**) Correlation heatmap between the key microbial phylotypes and metabolic parameters; (**B**) Visualization of the correlation network according to the correlation. Red nodes: the key microbial phylotypes altered by AAM administration; green nodes: the metabolic parameters; blue nodes: lipid metabolism and inflammatory response related genes in liver. The red lines and black lines represent positive and negative correlations, respectively. Line width indicates the strength of correlation. Only the significant edges were drawn in the network based on the correlation test (|r| > 0.6, FDR adjusted *p* < 0.05).

**Figure 9 foods-10-02436-f009:**
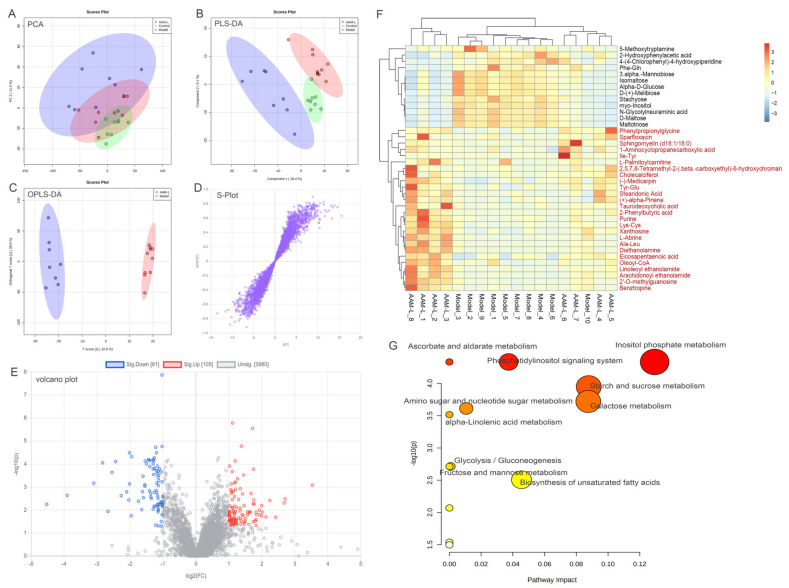
Liver metabolomic profiling by UPLC-QTOF/MS in the positive-ion mode (ESI+). (**A**) PCA score plot for the Control, Model and AAM-L groups; (**B**) PLS-DA score plot for the Control, Model and AAM-L groups; (**C**) OPLS-DA score plot for the AAM-L and Model groups; (**D**) S-loading plot based on the OPLS-DA analysis model of the Control and AAM-L groups; (**E**) Volcano plot of significantly different metabolites in liver (*p* < 0.05 and fold change > 2) between the Model and AAM-L groups; (**F**) Heatmap of the relative abundance of significantly different metabolites (VIP value > 1.0, *p* < 0.05 and fold change > 2) between the Model and AAM-L groups; (**G**) Metabolic pathway impact prediction based on the KEGG online database. The -ln(*p*) values from the pathway enrichment analysis are indicated on the horizontal axis, and the impact values are indicated on the vertical axis.

**Figure 10 foods-10-02436-f010:**
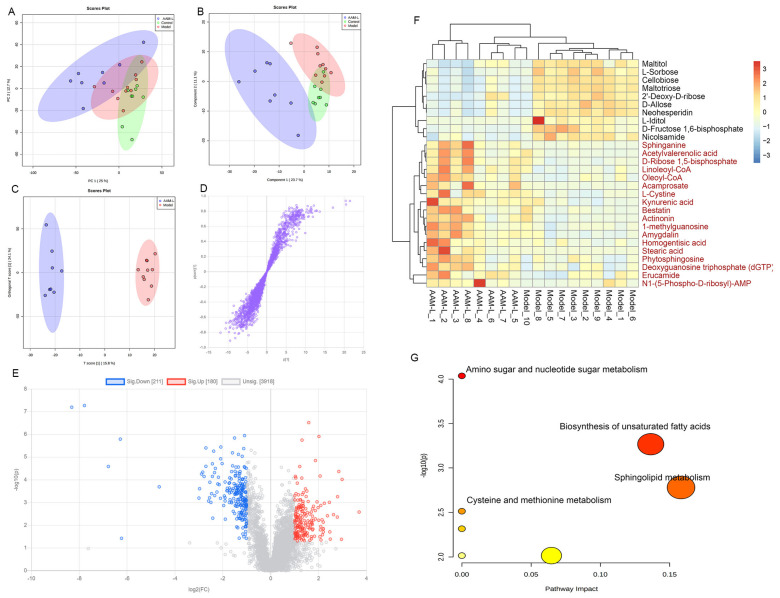
Liver metabolomic profiling by UPLC-QTOF/MS in the negative-ion mode (ESI-). (**A**) PCA score plot for the Control, Model and AAM-L groups; (**B**) PLS-DA score plot for the Control, Model and AAM-L groups; (**C**) OPLS-DA score plot for the AAM-L and Model groups; (**D**) S-loading plot based on the OPLS-DA analysis model of the Control and AAM-L groups; (**E**) Volcano Plot of significantly different metabolites between the Model and AAM-L groups; (**F**) Heatmap of relative abundance of significantly different metabolites (VIP value > 1.0, *p* < 0.05 and fold change > 2) between the Model and AAM-L groups; (**G**) Metabolic pathway impact prediction based on the KEGG online database. The -ln(*p*) values from the pathway enrichment analysis are indicated on the horizontal axis, and the impact values are indicated on the vertical axis.

**Table 1 foods-10-02436-t001:** Primer sequences for quantitative real-time PCR analysis.

Gene	Forward Primer (5′−3′)	Reverse Primer (5′−3′)
TNF-α	AAGCCTGTAGCCCACGTCGTA	AGGTACAACCCATCGGCTGG
LDLr	ATGCTGGAGATAGAGTGGAGTT	CCGCCAAGATCAAGAAAG
ACC1	GCCATCCGGTTTGTTGTCA	GGATACCTGCAGTTTGAGCCA
CPT-1	TCCATGCATACCAAAGTGGA	TGGTAGGAGAGCAGCACCTT
FASn	CTG CCA CAA CTC TGA GGA CA	TTC GTA CCT CCT TGG CAA AC
IFN-γ	ACAGCAAGGCGAAAAAGGATG	TGGTGGACCACTCGGATGA
CD36	ACTTGGGATTGGAGTGGTGATGT	GGATACCTGCAGTTTGAGCCA
18S	AGTCCCTGCCCTTTGTACACA	CGATCCCAGGGCCTCACTA

**Table 2 foods-10-02436-t002:** Elemental composition of *A. auricula* melanin.

Element Composition (%)					
Source	C (%)	H (%)	O (%) *	N (%)	S (%)
AAM	47.41	7.30	39.24	6.05	ND

* The oxygen content was calculated by the following equation: O% = 100% − C% − H% − N% − S%. ND indicates the element content was less than 0.5%.

## Data Availability

The data is kept in School of Research Center of JUNCAO Technology, Fujian Agriculture and Forestry University.

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
