# Peer review of "Auricularia auricula Melanin Protects against Alcoholic Liver Injury and Modulates Intestinal Microbiota Composition in Mice Exposed to Alcohol Intake"

_foods, 2021, doi:10.3390/foods10102436_

Round 1
Reviewer 1 Report
Auricularia auricula melanin protects against alcoholic liver injury and modulates intestinal microbiota composition in mice exposed to alcohol intake
I am writing you regarding manuscript # foods-1403464 entitled "Auricularia auricula melanin protects against alcoholic liver injury and modulates intestinal microbiota composition in mice exposed to alcohol intake," which you submitted to the foods.
The authors need to follow the following instructions to improve this manuscript.
- Page 1, Line 18-35 (Abstract section): Please, insert some relevant numeric results; and write the best findings
- Page 3, Line 104 (5000 ppm): convert to g value if possible. Follow the entire manuscript.
- Page 3, Line 106 (80℃): Make space between the number and degree centigrade symbol. Follow the entire manuscript.
- Page 3, Line 106 (hours): The authors used hours. Somewhere used h (line 176). Should follow same everywhere h.
- Figure 1: E should replace because it is not clear.
- Page 17, Line 747-760: Conclusions part should improve with the best findings.
- English Grammar should check by a Native English Speaker or commercial proofreading company.
- Reference writing style should check. Follow the author's guidelines.
- Please check carefully before resubmission.
I recommend improving the manuscript and resubmit.
Author Response
Q: Page 1, Line 18-35 (Abstract section): Please, insert some relevant numeric results; and write the best findings
A: Relevant findings and numeric results have been supplemented in the revised manuscript.
Q: Page 3, Line 104 (5000 ppm): convert to g value if possible. Follow the entire manuscript.
A: The centrifugal force unit in the full text has been converted from rpm to g.
Q: Page 3, Line 106 (80℃): Make space between the number and degree centigrade symbol. Follow the entire manuscript.
A: It has been changed as required in the entire manuscript.
Q: Page 3, Line 106 (hours): The authors used hours. Somewhere used h (line 176). Should follow same everywhere h.
A: It has been changed as required.
Q: Figure 1: E should replace because it is not clear.
A: The picture definition has been adjusted and replaced.
Q: Page 17, Line 747-760: Conclusions part should improve with the best findings.
A: Conclusions part has been improved with the best findings.
Q: English Grammar should check by a Native English Speaker or commercial proofreading company.
A: English grammar of the entire manuscript has been check by a Native English Speaker.
Q: Reference writing style should check. Follow the author's guidelines.
A: Reference writing style has been checked in the entire manuscript. The reference format has been modified.
Q: Please check carefully before resubmission.
A: We have check carefully before resubmission.
Reviewer 2 Report
I like the effort of the authors to evaluate the effect of supplementation with Auricularia auricula melanin in mice with alcoholic liver injury. In my opinion, they have carried out an extensive study with very interesting results. The beneficial effects reported suggest that AAM could be a novel food or functional component, although clinical studies are necessary to verify its effectiveness in humans.
In the following paragraphs, I will provide clear information in order to improve the manuscript. In general, the English level is good and I cannot see any discrepancies.
INTRODUCTION
The introduction is clear, fluid, and gives the necessary context for the article (Alcohol liver disease and the necessity to search for new natural compounds that may help in its prevention, A. auriculata properties, and bioactive compounds, and the relation between microbiota and long-term alcohol intake). The objectives have been also clearly exposed. Therefore, in my opinion, this section has been correctly performed. According to the authors, this is the first study evaluating the effects of Auricularia auricula melanin on the intestinal flora and liver metabolome of mice with alcoholic liver injury, so the study brings novelty to the scientific field.
MATERIAL AND METHODS
From my point of view, the scientific design and methodology employed are suitable to achieve the objectives proposed. In addition, the section has been described in a precise and clear way. I have only minor comments for the authors.
- Line 136: “normal group” should be “control group”. Please, correct throughout the manuscript.
- Line 157: The abbreviation of EDs is not mentioned in the text.
- Line 179:4°C. Please, correct.
- In subsection 2.9, I would suggest the authors separate "s" from the number. Ex. 15 s, 5 s, etc.
RESULTS & DISCUSSION
In my opinion, the authors have presented their numerous results in a clear, fluid and precise way and they have discussed them adequately. This section has been correctly performed. Tables present relevant information and the Figures are very informative, visual and easy to understand.
- Line 234: A. auriculata must be in italics.
CONCLUSIONS
Conclusions coincide with the results of the study and summarized them very well. In addition, the authors mention future steps of their research.
FINAL CONCLUSIONS
In my opinion, authors have performed an extensive work, with correct scientific bases and which provides interesting results for the food scientific field. Therefore, I am suggesting MINOR REVISIONS to improve its quality before publishing.
Author Response
Q: Line 136: “normal group” should be “control group”. Please, correct throughout the manuscript.
A: It has been changed as required throughout the manuscript.
Q: Line 157: The abbreviation of EDs is not mentioned in the text.
A: It has been changed as required.
Q: Line 179:4°C. Please, correct.
A: It has been changed as required.
Q: In subsection 2.9, I would suggest the authors separate "s" from the number. Ex. 15 s, 5 s, etc.
A: It has been changed as required.
Q: Line 234: A. auriculata must be in italics.
A: It has been changed as required.
Q: In my opinion, authors have performed an extensive work, with correct scientific bases and which provides interesting results for the food scientific field. Therefore, I am suggesting MINOR REVISIONS to improve its quality before publishing.
A: It has been changed as required.